# Disentangling the impact of Atlantic Niño on sea-air $CO_2$ flux

**Shunya Koseki** [1] ✉**, Jerry Tjiputra** [2]**, Filippa Fransner** [1]**, Lander R. Crespo** [1] **& Noel S. Keenlyside** [1,3]

Atlantic Niño is a major tropical interannual climate variability mode of the sea surface temperature (SST) that occurs during boreal summer and shares many similarities with the tropical Pacific El Niño. Although the tropical Atlantic is an important source of $CO_2$ to the atmosphere, the impact of Atlantic Niño on the sea-air $CO_2$ exchange is not well understood. Here we show that the Atlantic Niño enhances (weakens) $CO_2$ outgassing in the central (western) tropical Atlantic. In the western basin, freshwater-induced changes in surface salinity, which considerably modulate the surface ocean $CO_2$ partial pressure ($pCO_2$), are the primary driver for the observed $CO_2$ flux variations. In contrast, $pCO_2$ anomalies in the central basin are dominated by the SST-driven solubility change. This multi-variable mechanism for $pCO_2$ anomaly differs remarkably from the Pacific where the response is predominantly controlled by upwelling-induced dissolved inorganic carbon anomalies. The contrasting behavior is characterized by the high $CO_2$ buffering capacity in the Atlantic, where the subsurface water mass contains higher alkalinity than in the Pacific.

The tropical Atlantic and Pacific Oceans exhibit similarities in both climate and its interannual variability. Their eastern equatorial regions are characterized by strong upwelling[1,2] of cold subsurface water rich in dissolved inorganic carbon making the regions abundant sources of carbon to the atmosphere[3–5]. The two basins also experience similar equatorial pattern of SST variability modes that dominate at inter-annual timescale[2,6–9]: the Atlantic Niño and the El Niño/Southern Oscillation (ENSO) in the Pacific. These patterns of variability present two distinct phases that have impacts on climate and ecosystems both locally and globally[2,10,11] - the Atlantic Niño and El Niño are characterized by reduced upwelling and warmer SST, while the opposite occurs during Atlantic Niña and La Niña.

In contrast to the Pacific[3,4], knowledge is lacking on how this pattern of variability impacts the $CO_2$ fluxes in the equatorial Atlantic. Since the equatorial Atlantic represents a major $CO_2$ outgassing system and a region of early anthropogenic signal emergence in the global ocean[12], understanding how Atlantic Niño variability modulates the sea-air $CO_2$ fluxes is crucial to better constrain present and future

carbon budget and anthropogenic climate change. Here, we investigate how the $CO_2$ flux responds to the equatorial Atlantic interannual variability by identifying the $CO_2$ flux anomalies associated with the Atlantic Niño and compare them with the respective anomalies during El Niño in the equatorial Pacific, using observational SST and sea-air $CO_2$ flux data (see Methods).

Subsequently, we quantify the contribution of SST, sea surface salinity (SSS), dissolved inorganic carbon (DIC), and total alkalinity (ALK)[13,14] to driving the $CO_2$ partial pressure ($pCO_2$), and consequently the $CO_2$ flux anomalies in the equatorial Atlantic using observations and simulations of an IPCC-class model - the Norwegian Earth System Model version 2 (henceforth, NorESM[15,16]; this model was selected because of its good performance in the equatorial Atlantic and as it is one of the better performing CMIP6[17] ESMs in the equatorial Atlantic (see Method and Supplementary Table S1, Supplementary Figs. S1 and S2). For each driver of $pCO_2$, we elucidate its mechanism using NorESM2 because the usage of observational data of ocean carbon parameters like dissolved inorganic carbon and total alkalinity is

[1]Geophysical Institute, University of Bergen/Bjerknes Centre for Climate Research, Bergen, Norway. [2]NORCE Norwegian Research Centre/Bjerknes Centre for Climate Research, Bergen, Norway. [3]Nansen Environment and Remote Sensing Centre/Bjerknes Centre for Climate Research, Bergen, Norway. ✉e-mail: Shunya.Koseki@uib.no

limited. A brief model-data comparison of NorESM2 SST and sea-air $CO_2$ flux are given in Supplementary Fig. S1 and the Method section.

We show composite differences of governing parameters to emphasize the anomalies associated with the Atlantic Niño (i.e., the average difference between Atlantic Niño and Niña conditions; see Method). According to previous studies[8,18–21], the Pacific and Atlantic Niño variability have a few different mechanisms, patterns and periodicities (e.g., ENSO Modoki, canonical Atlantic Niño, and early/late onset of Atlantic Niños). We focus on the climatological peak of the Pacific and Atlantic Niños in November-December and June-July, respectively and investigate the most common $CO_2$ flux response to the mean climate variability modes without any detailed categorization of events.

## Results
### $CO_2$ flux anomalies in the tropical Pacific and Atlantic
The eastern equatorial Pacific is predominantly characterized by climatological cold SST, known as the cold tongue, and $CO_2$ outgassing, which are associated with the steady upwelling of cold and DIC-rich deep water[3–5] (Fig. 1a, 0.43 Pg C yr$^{-1}$ over the shown domain). The equatorial Atlantic is also a $CO_2$ outgassing system, but its spatial distribution does not closely follow the cold tongue, which is strongest from June to July (Fig. 1c, 0.075 Pg C yr$^{-1}$ over the shown domain). The spatial correlation coefficients between the observed climatological $CO_2$ fluxes and SST (Fig. 1a, c) are −0.69 and −0.29 (both are significant at 95% level) in the equatorial Pacific and Atlantic, respectively.

During the peak of El Niño phase from November to December, the $CO_2$ outflux is suppressed[3,4] in most parts of the central to eastern Pacific mainly due to less DIC upwelled from the deeper layers[3,4] (Fig. 1b, −0.270 Pg C yr$^{-1}$ area-integrated over the shown domain). Interestingly the $CO_2$ flux anomalies during Atlantic Niño differ remarkably from those in the Pacific: in the Atlantic, the strongest anomalies occur in the western basin away from the core of SST

anomalies in the central to eastern basin (Fig. 1d, −2.557 Tg C yr$^{-1}$ area-integrated over the shown domain), while the SST and $CO_2$ flux anomalies nearly co-located in the Pacific (Fig. 1b). In the western equatorial Atlantic, the $CO_2$ outgassing is suppressed during the Atlantic Niño, but the opposite pattern is observed in the central basin. The tendencies in the western Atlantic and in the central Pacific are similar, but they differ markedly in the east. The event selection of the Pacific and Atlantic Niño/Niña in this study is shown in Supplementary Fig. S3.

NorESM2 can reproduce the observed spatial peculiarities of sea-air $CO_2$ flux associated with the Atlantic Niño (Figs. 1d and 2b). This indicates that the model reasonably reproduces the mechanisms giving rise to the anomalous $CO_2$ fluxes, even though the Atlantic Niño tends to be overestimated (contours in Fig. 2 and Supplementary Fig. S1). In addition, the observation shows outgassing, while NorESM2 (as well as MRI-ESM2-0 and CESM2-WACCM in Supplementary Fig. S2b, c) shows ingassing, adjacent to the Amazon River mouth (2°N-6°N and 50°W-48°W). This might be because the observation-based $CO_2$ flux estimate we use has insufficient resolution to properly capturing the effect of Amazon River plume on $pCO_2$[22,23]. According to previous studies[24–26], the Amazon River plume is a large $CO_2$ sink around the Amazon River mouth and the influence of the plume extends eastward along the North Equatorial Countercurrent. In the NorESM2 simulation there is an extension of weak $CO_2$ sink and source to 40°W-35°W at 4°N to 6°N (Fig. 2a) and, to some extent, NorESM2 is able to capture the importance of the Amazon River plume in the western basin of the equatorial Atlantic. However, the $CO_2$ sink at 10°N is not well reproduced as in NorESM2 as in the previous observational study[25]. This is one of the limitations of the NorESM2. Because there is no observational data with sufficient spatial and temporal coverage required to investigate the drivers of $pCO_2$ change associated with the Atlantic Niño, the simulation of NorESM2 will be used to elucidate the driving mechanisms of the sea-air $CO_2$ flux anomalies.

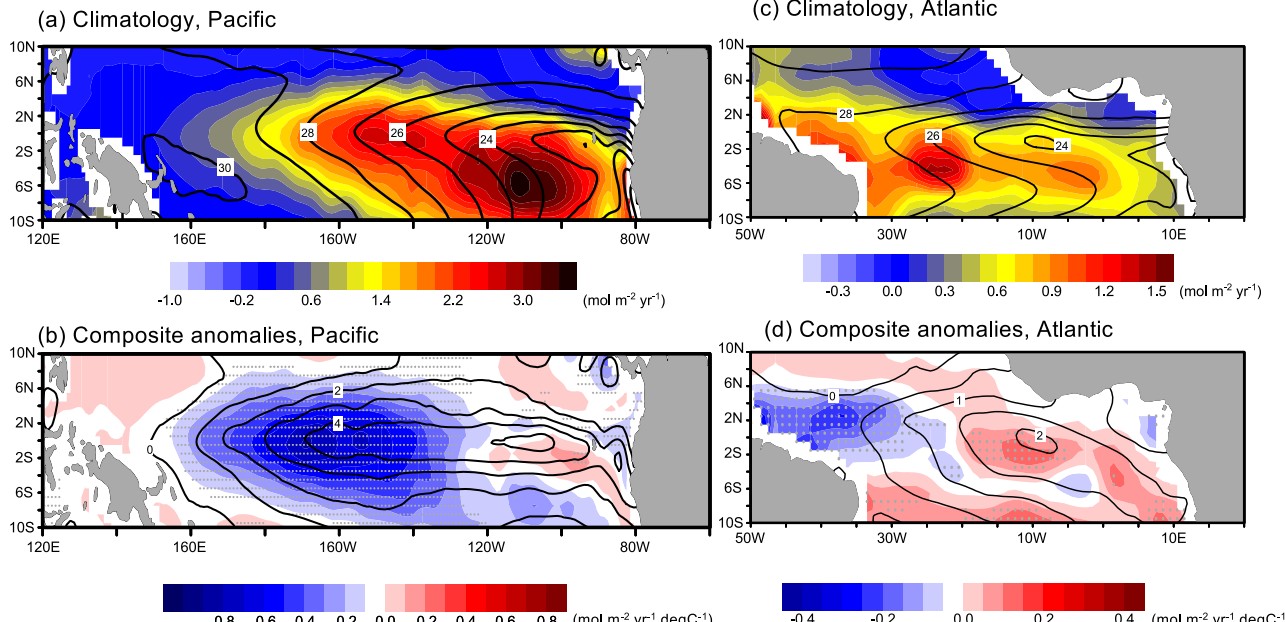

**Fig. 1 | Observed sea-air $CO_2$ flux and sea surface temperature in the tropical Pacific and Atlantic. a, c** Observed climatology (1990–2015) and (**b, d**) composite anomalies of sea surface temperature (SST, black contour, degree C) and sea-air $CO_2$ flux (color, mol m$^{-2}$ yr$^{-1}$) in the tropical Pacific (November-December) and Atlantic Ocean (June-July) between warm and cold events. Gray dots denote the air-sea carbon flux anomalies that are statistically significant at the 90% level estimated by the Student's $t$ test. Note that the sea-air carbon flux anomalies are scaled by NINO3.4 and Atlantic-3 indices anomalies (units in mol m$^{-2}$ yr$^{-1}$ degree C$^{-1}$) between Niño and Niña events for the Pacific and Atlantic, respectively. The observed data of $CO_2$ flux and SST are from MPI-SOM FFM[23,42] and OISST[41].

## Drivers of $p$CO$_2$ anomalies

The $p$CO$_2$ anomalies predominantly regulate the CO$_2$ flux patterns during Atlantic Niño (see the close resemblance in Fig. 2b and Supplementary Fig. S4b). On the other hand, the surface wind anomalies seem to be less responsible for forming the sea-air CO$_2$ flux patterns according to its climatology and anomaly distribution (Supplementary Fig. S5). To determine the governing mechanisms of the CO$_2$ flux anomalies, we decompose the $p$CO$_2$ anomalies into its main drivers: SST, SSS, ALK and DIC, applying the seawater inorganic carbonate system calculation (CO2SYS[27]) on the NorESM2 simulation.

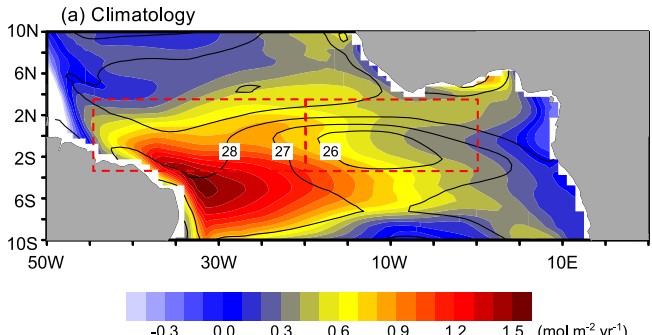

(a) Climatology

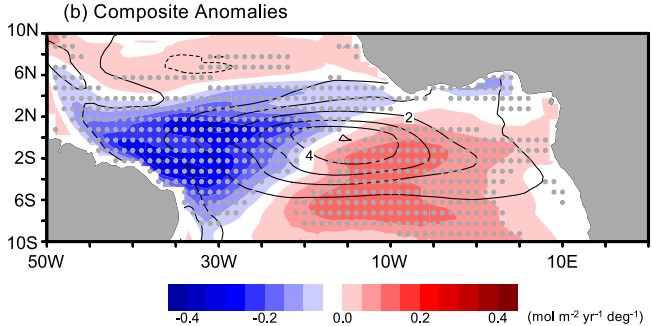

(b) Composite Anomalies

**Fig. 2 | Simulated sea-air CO$_2$ flux and sea surface temperature (SST).** Same as Fig. 1c, d, but for NorESM2 simulation, **a** climatology and **b** composite anomalies. Composite anomalies are scaled by the Atlantic-3 index composite anomaly between Niño and Niña events. The dashed red rectangles denote western and central basins defined in this study.

Supplementary Table S2 lists the combination of input variables for the CO2SYS calculation in order to decompose the relative contributions of each driver on the total $p$CO$_2$ anomaly. Here, the total $p$CO$_2$ anomaly ($p$CO$_{2F}$, where $F$ denotes four contributing variables) is compared with SST-, SSS-, ALK-, and DIC-driven $p$CO$_2$ anomalies (denoted as $p$CO$_{2T}$, $p$CO$_{2S}$, $p$CO$_{2A}$, and $p$CO$_{2D}$, respectively). The climatology and anomalies of CO2SYS-estimated $p$CO$_{2F}$ are nearly identical with the respective prognostic NorESM2 $p$CO$_2$ (Fig. 3a, b and Supplementary Fig. S5a, b). The climatological $p$CO$_{2F}$ has a maximum in the western basin between 4°S and the equator where the maximum CO$_2$ outflux occurs (Figs. 3a and 2a). Furthermore, the $p$CO$_{2F}$ anomalies also exhibit a consistent zonal dipole structure as seen in the sea-air CO$_2$ flux anomalies. This reaffirms that the spatial CO$_2$ flux anomaly pattern is, to a first order, governed mainly by the $p$CO$_2$ variability.

The positive SST anomaly during the Atlantic Niño (Fig. 2), induced by the reduced upwelling of cold water, has a positive effect on the $p$CO$_2$ anomaly through the reduced solubility associated with warmer surface waters (Fig. 3c). It reaches an amplitude of 85 µatm exceeding that of $p$CO$_{2F}$, and is found mainly in the central to eastern basin, consistent with the SST changes presented in the previous section. In addition, a weak and negative $p$CO$_{2T}$ anomaly is found in the northwestern basin, in particular off the equator.

During the Atlantic Niño, negative $p$CO$_{2S}$ anomalies, related to negative salinity (i.e., freshening) anomalies (Supplementary Fig. S6e, f), dominate the western basin with an amplitude of −25 µatm similar to the $p$CO$_{2F}$ anomalies (Fig. 3d). These negative salinity anomalies are induced mainly by anomalously higher precipitation in the large part of western basin (Supplementary Fig. S6a–d) during the Atlantic Niño. The high precipitation extends to the eastern equatorial Atlantic area[21] and NorESM2 simulations can capture its geographical distribution properly. Note that the precipitation anomaly associated with the Atlantic Niño is overestimated compared to the observation and its location shifts slightly southward (Supplementary Fig. S6b, d) partially because of the model's SST bias. In addition, a reduced advection of tropical high-saline waters (this advection is a crude estimate computed with monthly SSS and ocean current output) by the westward equatorial current[28] contributes to the lower salinity off the northern coast of the South American Continent. However, in NorESM2, the horizontal salinity advection anomalies are statistically less significant around the Amazon River mouth (Supplementary Fig. S6h). However, between 40°W-30°W at 2°N-4°N, the freshening due to the horizontal salinity

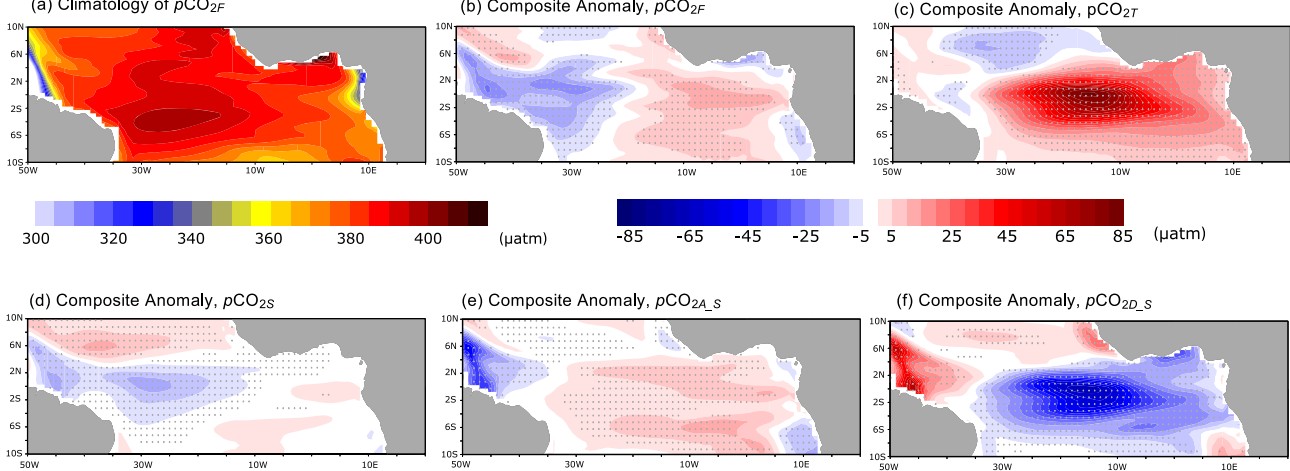

**Fig. 3 | Drivers of variability in partial pressure of CO$_2$.** **a** Climatology of partial pressure of CO$_2$ ($p$CO$_{2F}$) in June-July. **b–f** the composite difference of the different contributions to $p$CO$_2$ calculated between the Atlantic Niños and Niñas: **b** four variables, **c** sea surface temperature, **d** sea surface salinity, **e** salinity-normalized total alkalinity, and **f** salinity-normalized dissolved inorganic carbon, respectively. Gray dots denote 95% of significance level.

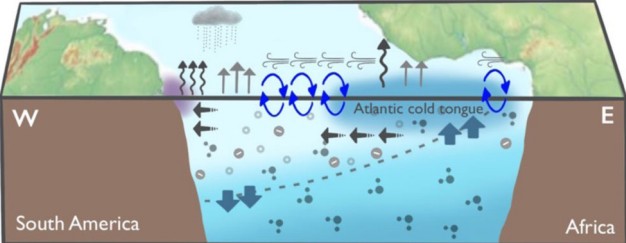

(a) Climatology

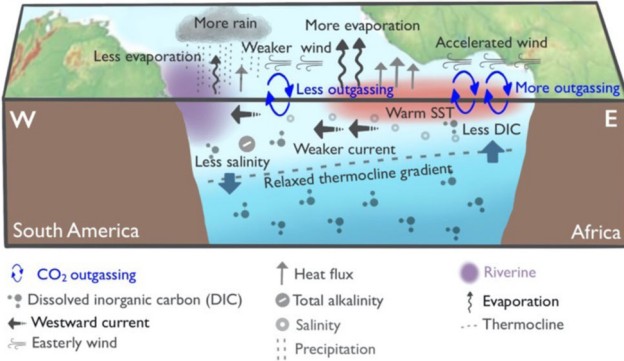

(b) Atlantic Niño

↻ CO₂ outgassing
•°• Dissolved inorganic carbon (DIC)
← Westward current
↝ Easterly wind

↑ Heat flux
⊘ Total alkalinity
○ Salinity
⋮ Precipitation

● Riverine
↕ Evaporation
--- Thermocline

**Fig. 4 | Schematic of the proposed mechanism of sea-air CO₂ flux response to Atlantic Niño.** Air-sea carbon flux of (**a**) climatology and (**b**) response to the Atlantic Niño suggested by this study.

advection is significant (Supplementary Fig. S6j) and therefore, there may contribute to the Amazon River plume in freshening the western basin until 30°W. This salinity advection anomaly is mainly due to the eastward anomalies of North Brazil Current and part of South Equatorial Current (Supplementary Fig. S6h, j). The North Equatorial Countercurrent anomalies are westward around 6°N (Supplementary Fig. S6h) in agreement with a previous study[29] and are therefore not responsible for freshening the western basin during Atlantic Niño events in our simulation. On the other hand, Amazon River discharge is reduced during Atlantic Niño event in NorESM2 and this does not contribute to the freshening anomaly (Supplementary Fig. S7). The potential contribution of Amazon river discharge and plume to interannual variability of $p\mathrm{CO}_2$ anomalies requires further analysis with higher resolution ocean models and observations. In the western basin, the negative latent heat flux anomalies also tend to freshen the ocean surface (Supplementary Fig. S6k–n). However, the positive latent heat flux anomaly in the central basin does not result in any positive SSS anomaly. Therefore, the latent heat flux is not a main driver for the salinity change in this region.

The contribution of carbonate species composition, ALK and DIC, to $p\mathrm{CO}_2$ is stronger than the SSS- and SST-induced solubility changes[30]. Similar to the $p\mathrm{CO}_{2S}$, the $p\mathrm{CO}_{2A}$ and $p\mathrm{CO}_{2D}$ anomalies dominate in the western basin (Supplementary Fig. S8a, b), indicating that upwelling- and freshwater-induced changes (through dilution and concentration) in surface ALK and DIC dominate the $p\mathrm{CO}_{2A}$ and $p\mathrm{CO}_{2D}$. As the ALK and DIC contributions to $p\mathrm{CO}_2$ change is almost identical with opposite sign[30], their associated contributions almost cancel out in the western basin. The $p\mathrm{CO}_2$ anomalies associated with salinity-normalized ALK ($p\mathrm{CO}_{2A\_S}$) and DIC ($p\mathrm{CO}_{2D\_S}$) more clearly show their contributions associated with the upwelling modification (Fig. 3e, f). The salinity-normalized anomalies (see Methods) exclude the role of surface freshwater fluxes and reflect the role of physical and biological activities on surface $p\mathrm{CO}_2$[30]. In the central basin, there is a weak positive $p\mathrm{CO}_{2A\_S}$ anomaly with an amplitude of 10 µatm that is statistically detectable, and a negative $p\mathrm{CO}_{2D\_S}$ anomaly with an amplitude of

70 µatm. The net effect of these anomalies is a reduction in the surface $p\mathrm{CO}_2$ during the Atlantic Niño.

The anomalies associated with $p\mathrm{CO}_{2A\_S}$ and $p\mathrm{CO2}_{D\_S}$ reflect the thermocline modulation associated with equatorial upwelling. In the subsurface between 60 and 80 m depth, there are ALK and DIC anomalies with a see-saw pattern from western to central/eastern basin, with the DIC anomaly being more remarkable (Supplementary Fig. S9). Because the vertical gradient of the DIC is much steeper than that of ALK (Supplementary Fig. 10a, c), the DIC is more sensitive to vertical displacement of the thermocline. The $p\mathrm{CO}_{2A\_S}$ and $p\mathrm{CO}_{2D\_S}$ show large anomalies in the western basin with opposite signs (Fig. 3e, f). As for the central basin, these anomalies can also be associated with the thermocline displacement. For example, during the Atlantic Niño, the climatological thermocline zonal tilt (shallower in the east) is reduced. Under this condition, the western basin experiences an upwelling anomaly and an eastward ocean surface current anomaly as indicated in Supplementary Figs. S9 and S6h, i. Consequently, the combined dynamical and biological impacts on the $p\mathrm{CO}_{2A\_S}$ and $p\mathrm{CO}_{2D\_S}$ result in a di-pole pattern (Fig. 3e, f). However, we note that the contribution of biological processes (photosynthesis and respiration) is acting in opposition to the upwelling-induced DIC change[31] and therefore, the $p\mathrm{CO}_{2D\_S}$ is dominated by the dynamical process. The anomalies in the western basin due to ALK and DIC approximately offset each other as indicated by $p\mathrm{CO}_{2AD}$ (Supplementary Fig. S8c).

## Discussions

Figure 4 summarizes the mechanism of sea-air CO₂ flux anomalies due to the Atlantic Niño deduced in this study (for the Atlantic Niña, the opposite pattern occurs). In the central equatorial Atlantic, the upwelling of DIC is reduced during an Atlantic Niño, similar to the equatorial Pacific. However, the combined ALK- and SST-induced $p\mathrm{CO}_2$ anomalies that also accompany the upwelling of the cold ALK-rich deep waters is larger than the DIC-induced $p\mathrm{CO}_2$ anomalies mainly due to the higher carbon buffer capacity in the Atlantic (as compared to the Pacific). Consequently, the prevailing $p\mathrm{CO}_2$ anomaly is positive during the Atlantic Niños. In the western Atlantic basin, the CO₂ flux and surface ocean $p\mathrm{CO}_2$ anomalies are opposite in sign to those in the central basin and are induced mainly by SSS changes. Additionally, the SST anomalies in the western basin during the Atlantic Niño tend to be opposite to that in the central basin even though its magnitude is considerably small. These opposing SST anomalies induce opposite surface $p\mathrm{CO}_2$ anomalies in the western basin to the central basin. The large-scale salinity-induced $p\mathrm{CO}_2$ anomaly is primarily attributed to the freshwater anomalies associated with the precipitation around the Brazilian coast. While it is less significant and limited in NorESM2, there is an indication that the horizontal advection of salinity modulated by the eastward anomaly of ocean current around the Amazon River mouth and in the western basin[25]. Further investigation on the role of the Amazon plume will be necessary.

The dominant role of SST and SSS in driving the sea-air CO₂ flux anomaly have been reported in a case study of warm SST anomaly event, observed from February to May in 2010, in the north tropical Atlantic, slightly to the north of the region in this study[32–34]. This warm SST anomaly event, however, is more related to the Atlantic Meridional Mode during spring while our investigated SST anomaly represents the Atlantic Zonal Mode during summer. Nevertheless a connection between these two climate modes through ocean dynamics is suggested[35].

While we identify Atlantic Niño events based on the climatological peak of variability and assume canonical dynamics, there are other mechanisms causing the Atlantic Niño[18–21]. As one non-canonical example, equatorial warm events form due to the advection of off-equatorial subsurface temperature anomalies from the north[19]. In these events the upwelled sea water would contain the chemical properties of the off-equatorial north Atlantic Ocean. This may lead to

different contributions of DIC and ALK to the $p$CO$_2$ anomalies than that analyzed in our more generalized Atlantic Niño.

Our results indicate that the Atlantic Niño and Pacific El Niño induce very different sea-air CO$_2$ flux anomalies (Fig. 1). This difference is most likely a result of different carbonate composition of the Pacific and Atlantic deep waters. Supplementary Fig. S10. The vertical structure of DIC and ALK in both the Pacific and Atlantic are to first order quite similar. This indicates that the upwelling-induced DIC and ALK contributions to $p$CO$_2$ anomaly would be similar in both basins. Different responses in the $p$CO$_2$ could result from differences in the carbon buffering capacity, which is proportional to the carbonate ion (CO$_3^{-2}$, which can be approximately ALK minus DIC[36]) concentration. The CO$_3^{-2}$ can be approximated from ALK minus DIC[36]. Supplementary Fig. S11 shows the surface carbonate ion concentration in observational data and as simulated by the NorESM2. The tropical Pacific has a lower buffer capacity than the tropical Atlantic, primarily due to the higher subsurface alkalinity and surface evaporation in the latter. Since the $p$CO$_2$ change due to DIC (d$p$CO$_2$/dDIC) is larger in the sea water with lower buffer capacity[36], the $p$CO$_2$ is more sensitive to the same DIC change in the tropical Pacific than in the tropical Atlantic. Supplementary Fig. S12 depicts the composite differences in CO$_2$ flux and each contribution to $p$CO$_2$ between Pacific El Niño and La Niña from NorESM2 simulation during November and December. In the central tropical Pacific, the salinity contribution is much less than the temperature (Supplementary Fig. S12b, c). Similar to the tropical Atlantic, the temperature anomaly enhances the upward CO$_2$ flux. The DIC contribution is much more dominant than the ALK contribution and the $p$CO$_{2D_S}$ anomaly (−180 μatm) is much more pronounced than that in the tropical Atlantic (−70 μatm, Fig. 3f). In the Pacific, the DIC contribution overwhelms the temperature contribution resulting in the reduction in CO$_2$ outflux at the equator (Supplementary Fig. S12a). On the other hand, the CO$_2$ flux anomaly is positive off the equator (Supplementary Fig. S12a), which is induced by the temperature. This positive anomaly is not seen in the observations and is common across CMIP6 models that simulate the observed negative anomaly in the equator[37]. In short, the CO$_2$ flux anomaly in the tropical Pacific can primarily be attributed to this DIC-driven $p$CO$_2$ change at the equator[3].

Here, we have revealed the unique pattern and underlying mechanism of CO$_2$ flux anomalies triggered by the Atlantic Niño, applying an observational-validated simulation from the latest generation of the Norwegian Earth System Model (NorESM2[15,16]). We showed that SSS and SST are the main drivers of the distinct anomaly pattern, and that upwelling of DIC only plays a secondary role. This pattern differs remarkably from that observed in the Pacific during ENSO events, where upwelling of DIC plays a major role. The smaller contribution of upwelling-induced DIC in the Atlantic compared to the Pacific can be attributed to the higher carbon buffer capacity of the Atlantic waters. These results indicate that, given the different governing mechanisms, the tropical Atlantic CO$_2$ flux responses to future climate change will also be different from that in the tropical Pacific[37].

This study focused on only Atlantic Niño events based on the geographically-fixed SST variability indices. However, as previous studies have shown, there are similar, but different types (also mechanisms) of the Niño events[18–21]. One of the desirable future studies is to investigate how different the CO$_2$ flux response is to the distinct types of the Atlantic Niños events, particularly when long term observational-based CO$_2$ flux estimates become available. Sustained long-term observations of salinity and water-column carbonate parameters would be crucial to reaffirm our understanding of how long-term climate change signals may affect the CO$_2$ flux variability.

According to recent studies, the Atlantic Niño variability in the CMIP5 models tends to be reduced under global warming[38–40]. This might imply that the CO$_2$ flux response to the Atlantic Niño variability can be modified in the warmer climate. As suggested in previous studies, the amplitude of SST anomaly associated with the Atlantic Niño

variability is reduced mainly due to less active SST-thermocline feedback[38] and suppressed surface wind variability[40]. In addition, the precipitation variability associated with the Atlantic Niño appears to be reduced[39]. All these changes would influence the $p$CO$_2$ and CO$_2$ flux and further investigation with inter-model comparison is needed.

## Methods
### Observational data and model
Optimum Interpolated Sea Surface Temperature[41] (OISST, 1990–2019) data is used for observational SST. For the sea-air CO$_2$ flux, a global observation-based gridded data[23,42] (1990–2015) is used. The observed data of ocean dissolved carbon species are obtained from GLODAP version 2[43]. The observed atmospheric data is from ERA5 reanalysis[44] for momentum flux, precipitation from GPCP[45], and latent heat flux from TropFlux[46]. The Norwegian Earth System Model version 2 (NorESM2[15,16]) is one of the state-of-the-art Earth system models that participate in the CMIP6. The configuration used in this study is 2 degree for the atmospheric and land surface components and 1 degree for the ocean and the sea-ice components (NorESM2-LM). The marine biogeochemical component, iHAMOCC[16], has the same resolution as the ocean component. In this study, we use the 3 ensemble-member data from 1990 to 2014 of the historical run. Because of large model spread and uncertainty, we need to assess the performance of NorESM2-LM comparing to other ESMs. Therefore, we additionally use 12 CMIP6 models: CESM2[47], CESM2-WACCM[48], CMCC-ESM2[49], CNRM-ESM2-1[50], GFDL-CM4[51], GFDL-ESM4[52], IPSL-CM6A-LR[53], MIROC-ES2L[54], MPI-ESM1-2-HR[55], MPI-ESM1-2-LR[56], MRI-ESM2-0[57], and UKESM1-0-LL[58] as summarized in Supplementary Table S1. The references for these 12 models are also given in Supplementary Table S1 for more details of model configurations.

### A brief model evaluation
Challenges in accurately reproducing the observed SST pattern in the equatorial Atlantic SST are common across state-of-the-art Earth system models[59–61]. As summarized Supplementary Table S1, the NorESM2 and 12 CMIP6 models are assessed in terms of SST and sea-air CO$_2$ flux. The statistical metrics for this assessment are root mean squared error and spatial correlation with respect to the observations. For climatology of SST and CO$_2$ flux and CO$_2$ flux anomaly, the statistical metrics are calculated over 6°S-2°N and 45°W-0 and 6°S-5°N and 45°W-10°E boxes, respectively. The box for the climatology covers the climatological peak of the Atlantic Cold Tongue and CO$_2$ outgassing and the box for CO$_2$ anomaly covers the entire di-pole anomaly. First we assess the capacity of 13 CMIP6 models in capturing the observed patterns of climatological SST, sea-air CO$_2$ flux, and their responses to the Atlantic Niño variability. As one of recent studies shows[60], the state-of-the-art ESMs still exhibit the large SST bias in the tropical Atlantic. However, there are also less biased models (i.e., low RMSE and high correlation with observations: NorESM2, CMCC-ESM2, MRI-ESM2-0, IPSL-CM6A-LR, and UKESM1-0-LL). These models with less SST bias models tend to better reproduce the climatology of sea-air CO$_2$ flux distribution (e.g., NorESM2 and MRI-ESM2-0; Supplementary Table S1). Both NorESM2 and MRI-ESM2-0 can also capture the CO$_2$ flux response to the Atlantic Niño variability. While UKESM1-0-LL is also able to capture the anomaly pattern well, its climatology of SST and CO$_2$ is relatively poorly simulated (Supplementary Table S1). NorESM2 and MRI-ESM2-0 can simulate the climatological CO$_2$ flux maximum in the western basin while reproducing more realistic Atlantic Cold Tongue (ACT, Fig. 2 and Supplementary Fig. S1). Contrastingly, the CO$_2$ flux maximum in CESM2-WACCM locates more eastward (Supplementary Fig. S1) and the development of ACT is not well captured. These SST and CO$_2$ biases are also seen in other poorly performing models (not shown). The CO$_2$ flux responses in MRI-ESM2-0 is also realistic with a di-pole structure along the equator. CESM2-WACCM has a mono-pole anomaly of less CO$_2$ outflux overlapping the warm SST anomaly along the

equator (Fig. Sd). This assessment illustrates the high uncertainty in $CO_2$ flux response to the Atlantic Niño variability simulated in CMIP6 models and indicates that models that better simulate the climatology of SST and $CO_2$ flux tend to reproduce more realistic di-pole structure of $CO_2$ response to Atlantic Niño events along the equator. NorESM2 is one of the best models and there are 3 ensemble members available for more robust results. This gives us in total 75 years, providing us the possibility of having more than 10 Atlantic Niño/Niña events, which is enough to construct canonical Atlantic Niño events.

According to this assessment, NorESM2 and MRI-ESM2-0 are the best models to reproduce the $CO_2$ flux climatology and its response to the Atlantic Niño variability. The inter-annual variability in the equatorial Atlantic is evaluated (Supplementary Fig. S1g). The observed variability is largest in June-to-July with a secondary peak in November to December. The NorESM2 and MRI-ESM2-0 are able to simulate successfully the summer peak in June-to-July, even though its amplitude is overestimated. The imperfect seasonality of variability is also a common issue among the state-of-the-art ESMs[62] and we consider the NorESM2 to be a good representative of CMIP6 models in terms of simulating the inter-annual climate variability in the tropical Atlantic.

### Definition of Atlantic Niños and Niña events
The June-July mean Atlantic-3 index is defined as the SST averaged over 20°W to 0° and 3°S and 3°N. The Atlantic Niño (Niña) event is defined when the detrended Atlantic-3 index is more (less) than plus (minus) one standard deviation of the Atlantic-3 index. For the model, the Atlantic Niño and Niña events have been defined by each ensemble member. All the composite plots are based on this selected Atlantic Niño and Niña events and their statistical significance are estimated by Student's $t$ test at the 90% of level. The anomalies of each variable are defined as the difference between the anomaly composite of Atlantic Niño and Atlantic Niña.

For the composite of $p$CO$_2$, the detrended data is also used to remove the long-term trend. The gray dots in Fig. 3 denote 95% of significance level estimated by Student's $t$-test.

### Decomposition of $p$CO$_2$ into drives
The sea-air $CO_2$ flux is governed by surface wind, the solubility parameter, and the $CO_2$ partial pressure ($p$CO$_2$) difference between ocean-atmosphere interface. The surface ocean $p$CO$_2$, in turn, is a function of the SST, SSS, DIC, and ALK. To quantify their contributions to the change in surface $p$CO$_2$, we utilized an open source MATLAB script of CO2SYS[27] that allows us to estimate the state of the carbonate system of oceanographic water samples and data. For the $p$CO$_2$ calculation, we used monthly SST, SSS, and surface concentrations of total alkalinity and dissolved inorganic carbon and climatology of silicate and phosphate as inputs. Note that, due to the nonlinearlity of the carbonate system, $p$CO$_{2F}$ is not the simple sum of $p$CO$_{2T}$, $p$CO$_{2S}$, $p$CO$_{2A}$, and $p$CO$_{2D}$.

### Definition of salinity-normalized total alkalinity and dissolved inorganic carbon
Normalization of total alkalinity (ALK) and dissolved inorganic carbon (DIC) with respect to salinity at the model surface layer is given by

$$\text{ALK}_s(x,y,t) = \text{ALK}(x,y,t) \times \text{SSS}_c(x,y)/\text{SSS}(x,y,t) \qquad (1)$$

$$\text{DIC}_s(x,y,t) = \text{DIC}(x,y,t) \times \text{SSS}_c(x,y)/\text{SSS}(x,y,t) \qquad (2)$$

Here, subscripts s and c denote salinity-normalized quantity and June-July climatology, respectively. $x$, $y$ and $t$ denote zonal and meridional grid point and time (June-July mean of each year), respectively.

### Calculation of salinity horizontal advection
The salinity horizontal advection in Supplementary Fig. S6 is estimated as follows,

$$\text{HADV} = -u\frac{\partial S}{\partial x} - v\frac{\partial S}{\partial y} \qquad (3)$$

here, HADV denotes the salinity horizontal advection. $u$, $v$, and $S$ are monthly outputs of NorESM2 for surface zonal current, meridional surface current, and sea surface salinity, respectively.

## Data availability
NorESM2-LM data used in this study have been deposited in the Zenodo database[63] (https://doi.org/10.5281/zenodo.7777376). CMIP6 data can be found at https://www.wcrp-climate.org/wgcm-cmip/wgcm-cmip6. ERA5 data can be found at https://cds.climate.copernicus.eu/#!/search?text=ERA5&type=dataset. GPCP data can be found at https://psl.noaa.gov/data/gridded/data.gpcp.html. OISST data can be found at https://www.ncei.noaa.gov/products/optimum-interpolation-sst. TropFlux data can be found at https://incois.gov.in/tropflux/. MPI SOM-FFN data can be found at https://www.ncei.noaa.gov/access/ocean-carbon-acidification-data-system/oceans/SPCO2_1982_present_ETH_SOM_FFN.html. GLODAP data can be found at https://www.glodap.info/.

## Code availability
The codes used in this study has been deposited deposited in the Zenodo database[63] (https://doi.org/10.5281/zenodo.7777376).

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

## Acknowledgements

This study is supported by the H2020 TRIATLAS project (grant# 817578; SK, JT, FF, LRC, NSK). J.T. acknowledges the Research Council of Norway funded project COLUMBIA (275268). The observational data of SST and $CO_2$ flux are available to download at https://www.ncei.noaa.gov/products/optimum-interpolation-sst and https://www.ncei.noaa.gov/access/ocean-carbon-data-system/oceans/MPI-ULB-SOM_FFN_clim.html, respectively. The observed DIC and ALK data are found at https://cdiac.ess-dive.lbl.gov/data/. The observed precipitation data is available at https://psl.noaa.gov/data/gridded/data.gpcp.html. The ERA5 reanalysis data is downloaded from https://www.ecmwf.int/en/forecasts/datasets/reanalysis-datasets/era5. The computational resources of NorESM simulation and data archive are supported by UNINETT Sigma2 AS (NN9039K and NS9560K). The schematic of Fig. 4 has been prepared by Jadelynn Fong.

## Author contributions

S.K. and J.T. have mainly investigated the partial pressure decomposition of $CO_2$ with F.F.'s comments. S.K., J.T., F.F., L.R.C., and N.S.K. have contributed to the analysis, interpretation and discussion of the results. S.K., J.T., F.F., L.R.C., and N.S.K. have also contributed to the manuscript development.

## Funding

## Competing interests

The authors declare no competing interest.
