## [Peer Review File · Nature Communications]

Disentangling the impact of Atlantic Niño on sea-air CO₂ fluxREVIEWER COMMENTS

Reviewer #1 (Remarks to the Author):

Review of 'Disentangling the impact of Atlantic Niño on sea-air CO₂ fluxes'

The manuscript submitted for review to Nature Communication investigates the impact of Atlantic Niño on air-sea CO₂ flux exchange. This is an important and noteworthy study – as the authors correctly note, understanding how Atlantic Niño variability modulates the air-sea CO₂ fluxes will help better constrain future carbon budget and anthropogenic climate change. But how does this study achieve that? That's exactly what is missing. Future projections are lacking, as is, an in-depth understanding of how different flavors of Atlantic Niños may result in a shift in the underlying mechanisms reported in this study. These and a couple of other issues are outlined below, which will require careful consideration and thought by the authors. Once these issues are addressed, the study may be considered for further review and publication in Nature Communications.

1. Reconciling the physical processes outlined in Figure 4 with existing understanding of Atlantic Niños – Vallès-Casanova et al. 2020 (GRL, <https://agupubs.onlinelibrary.wiley.com/doi/full/10.1029/2020GL087108>), and other references therein, have noted that during the peak of the Atlantic Niños (June-August), precipitation is enhanced over the West African Sub-Sahel region and eastern Atlantic, alongside western Atlantic and northeastern South America. The latter is captured in Figure 4. But there is no mention or reference to the higher precipitation in the eastern Atlantic sector. Whether the model simulations selected for this study correctly capture this enhanced precipitation, has important implications for isolating the drivers and their impact on air-sea CO₂ flux variability across the Atlantic basin. The authors also do not mention the Vallès-Casanova et al. 2020 study, which I found rather curious given that it is the most comprehensive study till date on Atlantic Niños.

2. Lines 83 – 86: At the very onset, the authors point out that due to a lack of observational data, the authors will rely on model simulation for this study. Indeed, the lack of observational data is a challenge, and is a real impediment towards obtaining an observational-based evidence of the driving mechanisms of the air-sea CO₂ flux anomalies during the Atlantic Niños. However, no model is perfect. This study should be ideally conducted in a multi-model framework. Even the authors do acknowledge this in Lines 187-188. The authors either need to run ensemble simulations with their NorESM2 model to capture uncertainty on their findings or do the same analysis with other models to present robust findings.

3. Finally, the study seems incomplete as it does not cover the full range of possibilities that may happen during an Atlantic Niño event (for example, early onset, early termination, late onset in many cases). The different mechanisms during each one of these cases should be easy to simulate within a model simulation framework. Neither did this study cover how understanding the drivers & mechanisms elucidated here will help understand how the tropical Atlantic will respond to climate change. For example, NorESM2 is part of the CMIP6 set of model simulations (I believe). Can the authors examine future projections to capture the response of Atlantic Niño-CO₂ flux relationship to future climate change? The authors should aspire to tackle either one of these, given that the manuscript has been submitted to Nature Communications.

4. A minor comment – the authors may want to include in more simple terms, calculation of tropical Atlantic air-sea CO₂ flux (in PgC/yr), how it compares to the tropical Pacific air-sea CO₂ flux (for example, see Ishii et al., 2014, <https://bg.copernicus.org/articles/11/709/2014/>) and the expected reduction in CO₂ outgassing during the peak of the Atlantic Niño season.

Reviewer #2 (Remarks to the Author):

The manuscript titled "Disentangling the impact of Atlantic Niño on sea-air CO₂ fluxes", submitted to Nature Communications, presents the analysis of different datasets and model results to disentangle the impact of the Atlantic Niño events over the sea-air CO₂ exchange in the tropical

Atlantic. Authors identify different mechanisms that differ from those observed in the tropical Pacific caused by the ENSO. They identify two different mechanisms affecting differently the western and central parts of the tropical Atlantic during Atlantic Niño events and propose explaining mechanisms of their results. Climate variability impact over the oceanic CO₂ sink is scarcely explored and thus, the topic of this manuscript focuses on a highly relevant issue concerning our understanding of the mechanisms governing sea-air CO₂ exchange. The manuscript is generally well written and informative. Nevertheless, I have two major concerns that I think they affect their results and interpretations and I feel should be solved before recommending its publication in Nature Communications.

The Atlantic Niño. The presentation of the Atlantic Niño and the ENSO variability and its identification for subsequent analysis lacks explanation. I believe this is partially due to length limitations but it is also the basis for the subsequent data analysis. Both events are presented as homogeneous and in the Atlantic, identified through SST anomalies during June-July. Nevertheless, these events are much more complex and, for instance, two different ENSO and Atlantic Niño patterns are being identified (see for instance Lübbecke et al., 2018; Richter and Tokinaga, 2021; Tyaquicã et al., 2017). These present different warming and cooling spatial patterns, affecting sea surface during both boreal summer and boreal spring: the canonical ENSO (Atlantic Niño) and the Modoki ENSO (Atlantic Niño). Authors should clarify this point in the text to avoid misinterpretations and better explain the restriction to June-July in their analysis. Also, the addition of a figure with the identified Atlantic Niños and Pacific ENSO years would be very helpful and would permit to compare their results regarding the identification of these events with other studies.

The western tropical Atlantic. Authors first use the interannual, gridded pCO₂ product created by Landschutzer et al. 2016 to identify general patterns of pCO₂ variability to then, explore the underlying drivers with a model. Using the gridded product of Landschutzer et al., they observe that, during the years identified as affected by Atlantic Niño events, the strongest pCO₂ anomalies occur in the western tropical Atlantic and do not match the observed SST anomalies in the basin. The climatology of Landschutzer et al. 2016 is known to heavily underrepresent the Amazon river plume, which during boreal summer, is transported by the NBC retroflection into the NECC and towards the central tropical Atlantic, occupying an area of nearly 1.5 million Km² and drastically changing the sea-air CO₂ exchange in this vast area. Thus, while the SST products used to compute SST anomalies are well established and certainly include the Amazon river plume, the lack of a precise representation of the impact of the Amazon river plume in the pCO₂ distribution may partially explain the unusual pattern identified in the western part of the basin. When using the model to disentangle the drivers of the identified patterns, from figures S4 and S5 seems clear that it also fails to represent the extent of the Amazon river plume and thus, their results on the entire western part of the basin should be taken with care. Authors should discuss how this misinterpretation of the large Amazon river plume affect the adequacy of their results and their interpretations. I suspect that this misrepresentation may partially explain the difficulties in interpreting their results in the western part of the basin.

References:

- Lübbecke, J. F., B. Rodríguez-Fonseca, I. Richter, M. Martín-Rey Marta, T. Losada, I. Polo, and N. S. Keenlyside, 2018: Equatorial Atlantic variability—Modes, mechanisms, and global teleconnections. *Wiley Interdiscip. Rev.: Climate Change*, 9, e527, <https://doi.org/10.1002/wcc.527>.
- Richter, I., and H. Tokinaga, 2021: The Atlantic Niño: Dynamics, thermodynamics, and teleconnections. *Tropical and Extra-Tropical Air-Sea Interactions*, S. K. Behera, Ed., Elsevier, 171-206.
- Tyaquicã, P., Veleza, D., Lefèvre, N., Araujo, M., Noriega, C., Caniaux, G., Servain, J., & Silva, T. (2017). Amazon Plume Salinity Response to Ocean Teleconnections. *Frontiers in Marine Science*, 4, 250. <https://doi.org/10.3389/fmars.2017.00250>

Reviewer #1 (Remarks to the Author):

Review of 'Disentangling the impact of Atlantic Niño on sea-air CO₂ fluxes'

The manuscript submitted for review to Nature Communication investigates the impact of Atlantic Niño on air-sea CO₂ flux exchange. This is an important and noteworthy study – as the authors correctly note, understanding how Atlantic Niño variability modulates the air-sea CO₂ fluxes will help better constrain future carbon budget and anthropogenic climate change. But how does this study achieve that? That's exactly what is missing. Future projections are lacking, as is, an in-depth understanding of how different flavors of Atlantic Niños may result in a shift in the underlying mechanisms reported in this study. These and a couple of other issues are outlined below, which will require careful consideration and thought by the authors. Once these issues are addressed, the study may be considered for further review and publication in Nature Communications.

We greatly appreciate the reviewer for his/her very useful and constructive comments and discussions on our manuscript. The primary focus of our study is to describe, and understand the mechanisms of sea-air CO₂ flux response to the “mean” Atlantic Niño in present climate. To our best knowledge, this has never been done before, and it is therefore an important piece lacking in our understanding of the global carbon budget.

We agree that an investigation of how different flavors of Atlantic Niños influence the sea-air CO₂ fluxes, and how the response would change in the future, would be important, but we think that this goes beyond the scope of the current paper. Regarding the point of future climate change, we are replying at the reviewer's comment #3 in more details showing an additional analysis (Figure R2). Please find more detailed responses under the separate points below, as well as responses to the other comments of the reviewer, along with how our manuscript has been revised. Please note that the modifications are shown in **red-color font** in the revised manuscript.

Please also note that we have moved the discussion on the difference in CO₂ flux response mechanism between the Pacific and Atlantic from Supplementary Information to the main body because this discussion is really important to highlight our new findings clearly. Please see lines 209-233.

1. Reconciling the physical processes outlined in Figure 4 with existing understanding of Atlantic Niños – Vallès-Casanova et al. 2020

(GRL, <https://agupubs.onlinelibrary.wiley.com/doi/full/10.1029/2020GL087108>), and other references therein, have noted that during the peak of the Atlantic Niños (June-August), precipitation is enhanced over the West African Sub-Sahel region and eastern Atlantic, alongside western Atlantic and northeastern South America. The latter is captured in Figure 4. But there is no mention or reference to the higher precipitation in the eastern Atlantic sector. Whether the model simulations selected for this study correctly capture this enhanced precipitation, has important implications for isolating the drivers and their impact on air-sea CO₂ flux variability across the Atlantic basin. The authors also do not mention the Vallès-Casanova et al. 2020 study, which I found rather curious given that it is the most comprehensive study till date on Atlantic Niños.

Thank you very much for this useful comment on precipitation response. We have added some description on the eastern equatorial Atlantic positive rainfall anomaly in the observations and NorESM (Fig. S6), while also referring to Vallès-Casanova et al. (2020). Please see lines 139-140.

2. Lines 83 – 86: At the very onset, the authors point out that due to a lack of observational data, the authors will rely on model simulation for this study. Indeed, the lack of observational data is a challenge, and is a real impediment towards obtaining an observational-based evidence of the driving mechanisms of the air-sea CO₂ flux anomalies during the Atlantic Niños. However, no model is perfect. This study should be ideally conducted in a multi-model framework. Even the authors do acknowledge this in Lines 187-188. The authors either need to run ensemble simulations with their NorESM2 model to capture uncertainty on their findings or do the same analysis with other models to present robust findings.

We agree with the reviewer that all models are imperfect, and that we did not pay enough attention to this in the first version of the manuscript. As Richter and Tokinaga (2020, *Climate Dynamics*, added to the reference list in the revised version) noted, the state-of-the-art CMIP6 models still exhibit large biases in the tropical Atlantic. In another study (Koseki et al., to be submitted), we demonstrate that NorESM configurations with less SST bias tend to have more realistic sea-air CO₂ flux. The NorESM2-LM that we use in this study is one of the better NorESM configurations with more realistic CO₂ climatology and response to the Atlantic Niño variability. This was our motivation for applying NorESM2-LM to investigate the mechanisms of sea-air CO₂ flux response in the tropical Atlantic. Please note that our study actually utilizes three ensemble members using the NorESM2-LM model as described in the Method section. We added more descriptions on it. Please see lines 309-312.

In addition to this, we concur with the reviewer that an assessment of the performance of CMIP6 models is highly valuable. We therefore assessed the CO₂ flux climatology and response to the Atlantic Niño variability in 12 additional CMIP6 models and summarized each model's statistical performance in Table S1. According to these statistical metrics, NorESM2 is one of the best models among the 13 (another well-performing model is MRI-ESM2-0). It becomes clearer that models with larger SST bias tend to poorly reproduce the CO₂ flux climatology and response (for example, CESM2-WACCM, CNRM-ESM2, and MPI-ESM1-2-LR as in Fig. S2b,c and Fig.R1). The plots for CNRM-ESM2 and MPI-ESM1-2-LR are added in this reply. In the revision, we have clarified our motivation for selecting NorESM2-LM, highlighting it as one of the few CMIP6 models to reproduce a realistic sea-air CO₂ flux climatology and its response to the Atlantic Niño variability, and therefore suitable for further analysis on the underlying mechanisms. We added this assessment analysis in the revised manuscript by adding new Fig.S1 and S2, and Table S1 and extended Method sections about model and model evaluation. Please see lines 276-281 and 285-312 for this CMIP6 model assessment in Method section.

Figure R1. SST (contour) and CO₂ flux (shading) climatology anomalies of CNRM-ESM2-1 and MPI-ESM1-2-LR.

3. Finally, the study seems incomplete as it does not cover the full range of possibilities that may happen during an Atlantic Niño (for example, early onset, early termination, late onset in many cases). The different mechanisms during each one of these cases should be easy to simulate within a model simulation framework. Neither did this study cover how understanding the drivers & mechanisms elucidated here will help understand how the tropical Atlantic will respond to climate change. For example, NorESM2 is part of the CMIP6 set of model simulations (I believe). Can the authors examine future projections to capture the response of Atlantic Niño-CO₂ flux relationship to future climate change? The authors should aspire to tackle either one of these, given that the manuscript has been submitted to Nature Communications.

Thank you very much for raising this important point. As pointed out by the reviewer, there are a few different mechanisms to govern the Atlantic Niño (e.g., Richter et al., 2013, *Nature Geosciences*) and several spatiotemporal patterns of Atlantic Niño suggested by Vallès-Casanova et al., 2020. However, the primary focus of this study is to mechanistically understand, and give a generalized picture, of how the CO₂ flux varies during an average Atlantic Niño. This is a first step that needs to be done before any further, more detailed, description on the spatiotemporal patterns can be done. Indeed, it would be possible to investigate how the difference flavors of Atlantic Niños affect the air-sea CO₂ fluxes in our model simulations. However, we cannot be sure that there are correctly represented due to the limited observational datasets of *p*CO₂ and its drivers (except for SST where we have a long satellite record). We therefore prefer to leave this question to future studies when

we have better observational constrains. In addition, it is very unsure if the model simulations can reproduce 4 categories of Atlantic Niño by Vallès-Casanova et al. (2020) and such categorization would need more simulations, models, and analyses. Therefore, we clarified our purpose of this study more in the Introduction and also added a brief discussion on importance of further categorization of the Atlantic Niño event referring to some literatures. Please see lines 71-75, 202-207, and 247-251.

Regarding the impact of climate change, we have added some descriptions referring to Crespo et al. (2022), Yang et al. (2022), and Worou et al. (2022). Please see lines 255-261. In addition, we deleted the phrases in the last part of abstract corresponding to climate change.

Figure R2 shows the CO₂ flux anomalies associated with the Atlantic Niño variability in NorESM2-LM. Basically, the CO₂ flux anomaly pattern appears under climate change as well, but there are some differences from that under present climate (Fig. 2): the outgassing anomaly in the central basin is more clearly and widely spreading in warmer climate than in current climate. The Atlantic Niño variability of the NorESM2-LM is reduced under climate change, consistent with Crespo et al. (2022) and Yang et al. (2022). This result will need to be assessed in an intermodal comparison framework because there are still uncertainties in models' SST climatology, SST response to climate change and CO₂ flux processes. Therefore, we do not go deeply into details about CO₂ flux response to climate change in this study.

Figure R2. CO₂ flux anomalies of NorESM2-LM between Atlantic Niño and Niña composite during 2074-2099 scaled by Atlantic-3 Index anomaly amplitude.

4. A minor comment – the authors may want to include in more simple terms, calculation of tropical Atlantic air-sea CO₂ flux (in PgC/yr mol m⁻² yr⁻¹), how it compares to the tropical Pacific air-sea CO₂ flux (for example, see Ishii et al., 2014, <https://bg.copernicus.org/articles/11/709/2014/>) and the expected reduction in CO₂ outgassing during the peak of the Atlantic Niño season.

Thank you very much for suggesting a comparison in more common unit of CO₂ flux. For the observation, we calculated area-aggregated value of CO₂ anomalies over the tropical Atlantic (2S-6N/50W-25W) and Pacific (10S-8N/180W-120W) to compare the reduction of CO₂ flux due to the Niño event. According to this calculation, the tropical Atlantic has 5.955 Tg C yr⁻¹ and the tropical Pacific has 0.1995 Pg C yr⁻¹. This large difference is of course due to the difference in basin area and larger amplitude of the events in the Pacific. We have added these values and corresponding statement in the revised manuscript. Please see line 80, 82, 88, and 91.

Reviewer #2 (Remarks to the Author):

The manuscript titled “Disentangling the impact of Atlantic Niño on sea-air CO₂ fluxes”, submitted to Nature Communications, presents the analysis of different datasets and model results to disentangle the impact of the Atlantic Niño events over the sea-air CO₂ exchange in the tropical Atlantic. Authors identify different mechanisms that differ from those observed in the tropical Pacific caused by the ENSO. They identify two different mechanisms affecting differently the western and central parts of the tropical Atlantic during Atlantic Niño events and propose explaining mechanisms of their results. Climate variability impact over the oceanic CO₂ sink is scarcely explored and thus, the topic of this manuscript focuses on a highly relevant issue concerning our understanding of the mechanisms governing sea-air CO₂ exchange. The manuscript is generally well written and informative. Nevertheless, I have two major concerns that I think they affect their results and interpretations and I feel should be solved before recommending its publication in Nature Communications.

We greatly appreciate the reviewer for his/her very encouraging and constructive comments and discussions on our manuscript.

As below, we have replied point-by point to the reviewer’s constructive comments and modified our manuscript by adding some new figures and analysis that has been suggested by the other reviewer as well. Please note that the modifications are shown by **red-color font** in the revised manuscript.

We have modified much of the texts by adding new supplemental figures following the other reviewer’s comments. For assessment of model’s accuracy, we have added an analysis on CO₂ flux anomalies from other 12 CMIP6 models (please see Method section and new Table S1 and new Figure S1 and S2). Based on this new analysis show that NorESM2 is one of the best performing models among 13 in the tropical Atlantic.

We have moved the discussion on difference in the CO₂ flux response mechanism between the Pacific and Atlantic from Supplementary Information of the original version to the main body of the revised manuscript because this discussion is really important to highlight our new findings clearly. Please see lines 209-233.

The Atlantic Niño. The presentation of the Atlantic Niño and the ENSO variability and its identification for subsequent analysis lacks explanation. I believe this is partially due to length limitations but it is also the basis for the subsequent data analysis. Both events are presented as homogeneous and in the Atlantic, identified through SST anomalies during June-July. Nevertheless, these events are much more complex and, for instance, two different ENSO and Atlantic Niño patterns are being identified (see for instance Lübbecke et al., 2018; Richter and Tokinaga, 2021; Tyaquiçã et al., 2017). These present different warming and cooling spatial patterns, affecting sea surface during both boreal summer and boreal spring: the canonical ENSO (Atlantic Niño) and the Modoki ENSO (Atlantic Niño). Authors should clarify this point in the text to avoid misinterpretations and better explain the restriction to June-July in their analysis. Also, the addition of a figure with the identified Atlantic Niños and Pacific ENSO years would be very helpful and would permit to compare their results regarding the identification of these events with other studies.

Thank you very much for raising this important aspect of the Niño events. We totally agree that more investigation on different types of the Niño events will be needed. On the other hand, in this study we are focusing on the climatological peak of the Atlantic Niño (in general in June to July) and we decided to investigate “general” cases of the events without any further classification. We believe this is appropriate for a first study on this topic. In addition, it is unsure how many different events can be captured in our model sets. We believe that our findings can be a good milestone for further investigations on CO₂ flux response in different types of the Niño events since this study is the first study to reveal the CO₂ flux response. We have added more descriptions on our fundamental philosophy, the necessity of further investigation based on multiple aspects of the events and brief discussion on one example of non-canonical type of Atlantic Niño referring to Richter et al. (2013). Please see lines 71-75, 202-207, and 247-251.

Regarding the event identifications, we have added a new Fig.S3 to show the time series of the observed NINO3.4 and Atlantic-3 indices in this study. Please see the new Fig.S3 and lines 95. Please note that since we have used Earth system models, which simulate their own internal variability, the timing of Atlantic Niño events differs from that of the observations.

The western tropical Atlantic. Authors first use the interannual, gridded pCO₂ product created by Landschutzer et al. 2016 to identify general patterns of pCO₂ variability to then, explore the underlying drivers with a model. Using the gridded product of Landschutzer et al., they observe that, during the years identified as affected by Atlantic Niño events, the strongest pCO₂ anomalies occur in the western tropical Atlantic and do not match the observed SST anomalies in the basin. The climatology of Landschutzer et al. 2016 is known to heavily underrepresent the Amazon river plume, which during boreal summer, is transported by the NBC retroreflection into the NECC and towards the central tropical Atlantic, occupying an area of nearly 1.5 million Km² and drastically changing the sea-air CO₂ exchange in this vast area. Thus, while the SST products used to compute SST anomalies are well established and certainly include the Amazon river plume, the lack of a precise representation of the impact of the Amazon river plume in the pCO₂ distribution may partially explain the unusual pattern identified in the western part of the basin. When using the model to disentangle the drivers of the identified patterns, from figures S4 and S5 seems clear that it also fails to represent the extent of the Amazon river plume and thus, their results on the entire western part of the basin should be taken with care. Authors should discuss how this misinterpretation of the large Amazon river plume affect the adequacy of their results and their interpretations. I suspect that this misrepresentation may partially explain the difficulties in interpreting their results in the western part of the basin.

Thank you very much for raising this important detail. Even in the climatology, the Landschützer et al and modeled CO₂ fluxes are quite different around the Amazon River mouth, 2N-6N and 50W-45W (Figs. 1 and 2): CO₂ flux is negative in the model and the observation is positive. We would think that this discrepancy might be related to the uncertainty in the data product as the reviewer raised and we agree that it should be mentioned in the manuscript. Also, we found one paper, Coles et al. (The pathways and properties of the Amazon River Plume in the tropical North Atlantic Ocean, 2013, *JGR-Oceans*) in which the authors claim that the seasonal cycle of salinity in the tropical North Atlantic is affected equally by freshwater from precipitation and river plume through the ocean currents shown by PIRATA mooring data and a relatively high-resolution (1/6 degree) model. This conclusion is exactly what the reviewer points out. As Fig.S6l shows, the eastward current is intensified during the Atlantic Niño, partially due to the eastward surface wind anomalies associated with the Atlantic Niño (there would be some uncertainty though) and the river plume water could extend more eastward.

Regarding this point, we have added another Supplemental Figure S6i and S6j (please note that the order of Fig.S6 has been changed in the revised version) to show the salinity horizontal advection in

the western basin (due to the color scale emphasizing the value around the river mouth, Fig.S6h does not show clearly the value in the western basin). Figure S6j showed the freshening at 4N to 6N overlapping with the eastward current anomaly and indicating the eastward extension of the river plume. During the Atlantic Niño, the large part of western basin is freshened by the salinity horizontal advection with eastward current anomaly. It is unsure how realistic this current anomaly is, but as the reviewer and Coles et al. (2013), the salinity is affected by the river plume in the western basin.

Please note that the salinity advection is calculated from monthly SSS and ocean current data due to the data limitation. Therefore, the transient/residual component (daily scale) is lacking. We failed to mention this point in the submitted version and added it in the caption of Fig.S6 and Method Section (please see lines 360-365).

We added these discussions in the revised manuscript. Please see lines 143-150.

References:

- Lübbecke, J. F., B. Rodríguez-Fonseca, I. Richter, M. Martín-Rey Marta, T. Losada, I. Polo, and N. S. Keenlyside, 2018: Equatorial Atlantic variability—Modes, mechanisms, and global teleconnections. *Wiley Interdiscip. Rev.: Climate Change*, 9, e527, <https://doi.org/10.1002/wcc.527>.
- Richter, I., and H. Tokinaga, 2021: The Atlantic Niño: Dynamics, thermodynamics, and teleconnections. *Tropical and Extra-Tropical Air-Sea Interactions*, S. K. Behera, Ed., Elsevier, 171-206.
- Tyaquiçã, P., Veleza, D., Lefèvre, N., Araujo, M., Noriega, C., Caniaux, G., Servain, J., & Silva, T. (2017). Amazon Plume Salinity Response to Ocean Teleconnections. *Frontiers in Marine Science*, 4, 250. <https://doi.org/10.3389/fmars.2017.00250>

Thank you very much for suggesting further references. We added these papers to citation.

REVIEWER COMMENTS

Reviewer #1 (Remarks to the Author):

The revised version of this manuscript is significantly improved (relative to the original submission) and provides a much more comprehensive and detailed overview of the impact of Atlantic Niño on air-sea CO₂ flux exchange. The authors have addressed deficiencies identified in the original manuscript and provided more quantitative characterization of their results by conducting new analysis and simulations.

I highly recommend the study be published in Nature Communications. It provides a novel and unique perspective on how Atlantic Niño variability modulates air-sea CO₂ fluxes and lays the foundation for future studies and research from the geosciences community on this topic.

Reviewer #2 (Remarks to the Author):

The revised manuscript titled Disentangling the impact of Atlantic Niño on sea-air CO₂ flux, submitted to Nature Communications, presents several changes following the reviewers' comments that have improved its presentation compared to the original draft. Nevertheless, I still think authors should better place their valuable study within what is currently known about the impact of Atlantic Niño events over sea-air CO₂ exchange, the limitations of their modeling study and thus, the generalization of their results.

The limitations on the western part of the basin caused by an incomplete account of the Amazon River plume as pointed in the previous revision are still not properly addressed. The authors state in the manuscript:

Line 105: "However, this only affects a small part of our domain."

Lines 147-150: "Nevertheless, observational data and relatively high resolution model output indicate that Amazon river discharge and plume contribute substantially to the seasonal cycle of salinity in the tropical Atlantic (25). The potential contribution of Amazon river discharge and plume to interannual variability of pCO₂ anomalies requires further analysis with higher resolution ocean models and observations."

Lines 198-200: "While it is less significant and limited in NorESM2, there is an indication that the horizontal advection of salinity modulated by the westward ocean current around the Amazon River mouth. Further investigation on the role of the Amazon plume will be necessary."

The surface currents in the vicinity of the Amazon River plume present strong seasonality, with the reinforce of the North Equatorial Countercurrent (NECC) during the summer that advects Amazon waters to the East and towards the central Tropical Atlantic. This advection extends the plume to an area of 1.5 Km², which I believe is not "a small part of our domain". This occurs during May to September and thus, their model fails to reproduce the surface circulation (NECC) as well, since they show westward salinity advection instead of eastward. The authors present a modeling exercise and as such, there are limitations, but these should be presented and discussed. This is particularly relevant in their study as the authors attempt to explain their contrasting west-to-east patterns while neglecting the limitations of their model. Currently, there is already a very decent piece of observational evidence on the dynamics of the Amazon River plume. See, for instance, these relatively recent references as examples, apart from those cited in the manuscript:

Körtzinger, A., 2010. The outer Amazon plume: An atmospheric CO₂ sink, in: Liu, K.-K., Atkinson, L., Quiñones, R., Talaue-McManus, L. (Eds.), Carbon and Nutrient Fluxes in Continental Margins: A Global Synthesis. Springer, New York, USA, pp. 450–453.

Monteiro, T., Batista, M., Henley, S., Machado, E. da C., Araujo, M., Kerr, R., 2022. Contrasting Sea-Air CO₂ Exchanges in the Western Tropical Atlantic Ocean. *Global Biogeochemical Cycles* 36, e2022GB007385. <https://doi.org/10.1029/2022GB007385>

Mu, L., Gomes, H. do R., Burns, S.M., Goes, J.I., Coles, V.J., Rezende, C.E., Thompson, F.L., Moura, R.L., Page, B., Yager, P.L., 2021. Temporal Variability of Air-Sea CO₂ flux in the Western Tropical North Atlantic Influenced by the Amazon River Plume. *Global Biogeochemical Cycles* 35, e2020GB006798. <https://doi.org/10.1029/2020GB006798>

Both in the responses to the reviewers' comments and in the main text, the authors claim their

study is the first one partially unveiling the impact of the Atlantic Niños over sea-air CO₂ fluxes in the basin:

Response to reviewers: "We believe that our findings can be a good milestone for further investigations on CO₂ flux response in different types of the Niño events since this study is the first study to reveal the CO₂ flux response"

Lines 71-75 in the manuscript: "According to previous studies (8, 17-20), the Pacific and Atlantic Niños variability have a few different mechanisms, patterns and periodicities (e.g., ENSO Modoki, canonical Atlantic Niño, and early/late onset of Atlantic Niños). As a first study, we focus on the climatological peak of the Pacific and Atlantic Niños in November-December and June-July, respectively and investigate the most common CO₂ flux response to the mean climate variability modes without any detailed categorization of events."

Lines 105-108: "Because there is no observational data with sufficient spatial and temporal coverage required to investigate the drivers of pCO₂ change associated with the Atlantic Niño, the simulation of NorESM2 will be used to elucidate the driving mechanisms of the sea-air CO₂ flux anomalies."

Lines 237-238: "Here, we have, for the first time, revealed the unique pattern and underlying mechanism of CO₂ flux anomalies triggered by the Atlantic Niño"

While I recognize the value of this study, the authors should be aware of that three previous observational-based studies have studied the impact of an Atlantic Niño event over the sea-air CO₂ exchange in the basin:

Lefèvre, N., Caniaux, G., Janicot, S., Gueye, A.K., 2013. Increased CO₂ outgassing in February-May 2010 in the tropical Atlantic following the 2009 Pacific El Niño. *Journal of Geophysical Research: Oceans* 118, 1645–1657. <https://doi.org/10.1002/jgrc.20107>

Ibáñez, J.S.P., Flores, M., Lefèvre, N., 2017. Collapse of the tropical and subtropical North Atlantic CO₂ sink in boreal spring of 2010. *Scientific Reports* 7, 41694. <https://doi.org/10.1038/srep41694>

Lefèvre, N., Veleza, D., Tyaquicã, P., Perruche, C., Diverrès, D., Ibáñez, J.S.P., 2019. Basin-Scale Estimate of the Sea-Air CO₂ Flux During the 2010 Warm Event in the Tropical North Atlantic. *Journal of Geophysical Research: Biogeosciences* 124, 973–986. <https://doi.org/10.1029/2018JG004840>

The domain of these studies is different to that presented here and they are limited to a particular event, but they offer valuable observation on factors that I feel are not properly addressed by their model such as the abnormal displacement of the ITCZ during this event (Lefèvre et al., 2013), identifying warming as the main driver of the CO₂ anomalies observed (Ibáñez et al., 2017) and quantifying the impact over the sea-air CO₂ exchange in the basin north to the equator (Ibáñez et al., 2017, Lefèvre et al., 2019).

Other apparent limitations based on the results presented (for instance accounting for the anomalous displacement of the ITCZ, never mentioned in the manuscript) makes me feel that authors should improve the discussion of the limitations of the model while further mild the conclusions taken from their modeling results.

Reviewer #1 (Remarks to the Author):

The revised version of this manuscript is significantly improved (relative to the original submission) and provides a much more comprehensive and detailed overview of the impact of Atlantic Niño on air-sea CO₂ flux exchange. The authors have addressed deficiencies identified in the original manuscript and provided more quantitative characterization of their results by conducting new analysis and simulations.

I highly recommend the study be published in Nature Communications. It provides a novel and unique perspective on how Atlantic Niño variability modulates air-sea CO₂ fluxes and lays the foundation for future studies and research from the geosciences community on this topic.

REPLY: Thank you very much for the reviewer's constructive comments for the previous revision and our manuscript has been improved significantly.

Reviewer #2 (Remarks to the Author):

The revised manuscript titled Disentangling the impact of Atlantic Niño on sea-air CO₂ flux, submitted to Nature Communications, presents several changes following the reviewers' comments that have improved its presentation compared to the original draft. Nevertheless, I still think authors should better place their valuable study within what is currently known about the impact of Atlantic Niño events over sea-air CO₂ exchange, the limitations of their modeling study and thus, the generalization of their results.

REPLY: Thank you very much for the reviewer's constructive comments. As below, we have replied point-by-point and revised our manuscript adding more discussions. Please note that all the corrections in the revised manuscript are shown by red-color font.

The limitations on the western part of the basin caused by an incomplete account of the Amazon River plume as pointed in the previous revision are still not properly addressed. The authors state in the manuscript:

Line 105: "However, this only affects a small part of our domain."

Lines 147-150: "Nevertheless, observational data and relatively high resolution model output indicate that Amazon river discharge and plume contribute substantially to the seasonal cycle of salinity in the tropical Atlantic (25). The potential contribution of Amazon river discharge and plume to interannual variability of pCO₂ anomalies requires further analysis with higher resolution ocean models and observations."

Lines 198-200: "While it is less significant and limited in NorESM2, there is an indication that the horizontal advection of salinity modulated by the westward ocean current around the Amazon River mouth. Further investigation on the role of the Amazon plume will be necessary."

The surface currents in the vicinity of the Amazon River plume present strong seasonality, with the reinforce of the North Equatorial Countercurrent (NECC) during the summer that advects Amazon waters to the East and towards the central Tropical Atlantic. This advection extends the plume to an area of 1.5 Km², which I believe is not “a small part of our domain”. This occurs during May to September and thus, their model fails to reproduce the surface circulation (NECC) as well, since they show westward salinity advection instead of eastward. The authors present a modeling exercise and as such, there are limitations, but these should be presented and discussed. This is particularly relevant in their study as the authors attempt to explain their contrasting west-to-east patterns while neglecting the limitations of their model. Currently, there is already a very decent piece of observational evidence on the dynamics of the Amazon River plume. See, for instance, these relatively recent references as examples, apart from those cited in the manuscript:

Körtzinger, A., 2010. The outer Amazon plume: An atmospheric CO₂ sink, in: Liu, K.-K., Atkinson, L., Quiñones, R., Talaue-McManus, L. (Eds.), *Carbon and Nutrient Fluxes in Continental Margins: A Global Synthesis*. Springer, New York, USA, pp. 450–453.

Monteiro, T., Batista, M., Henley, S., Machado, E. da C., Araujo, M., Kerr, R., 2022. Contrasting Sea-Air CO₂ Exchanges in the Western Tropical Atlantic Ocean. *Global Biogeochemical Cycles* 36, e2022GB007385. <https://doi.org/10.1029/2022GB007385>

Mu, L., Gomes, H. do R., Burns, S.M., Goes, J.I., Coles, V.J., Rezende, C.E., Thompson, F.L., Moura, R.L., Page, B., Yager, P.L., 2021. Temporal Variability of Air-Sea CO₂ flux in the Western Tropical North Atlantic Influenced by the Amazon River Plume. *Global Biogeochemical Cycles* 35, e2020GB006798. <https://doi.org/10.1029/2020GB006798>

REPLY: Thank you very much for providing more discussion and information on the importance of role of the Amazon River plume in sea-air CO₂ flux. First, we looked more carefully the North Equatorial Countercurrent (NECC) and CO₂ flux in our model with referring to the observational evidences (Monteiro et al., 2022; Mu et al., 2021). As shown in Fig.2a, the simulated climatological sea-air CO₂ flux is negative around the Amazon River mouth (Eq-10N and 50W), which we have not mentioned in the previous revised manuscript. This uptake of CO₂ by the ocean is, to some extent, consistent with Fig. 2b of Monteiro et al. (2022).

In addition, the weakly negative and positive CO₂ flux extends eastward until 35W around 6N in NorESM2 as shown in Fig.2a. This indicates that the fresh water from the Amazon River plume is transported by the ocean current. Looking at Fig.S6g and i, the eastward current, representing be the NECC, is simulated well around 6N and freshening occurs between 40W and 25W (please note that the purplish color is very faint because the positive value is highlighted in the figure). Therefore, to some extent, NorESM2 is able to reproduce the influence of the Amazon River plume on the sea-air CO₂ flux in the western basin, however comparing to, for example, Monteiro et al. (2022), the area of CO₂ sink in the NorESM2 simulation is limited between 50W and 40W and this is one of the shortcomings in our simulation.

During the Atlantic Niño, the freshening occurs associated with the eastward extension of the Amazon River plume as shown in Fig.S6j, however, in our simulation, this extension is induced by the anomalies of North Brazilian Current and Equatorial Current, not by the NECC. According to Hormann et al. (2012), the NECC tends to be weakened during the Atlantic Niño (please see Fig.11d, <https://agupubs.onlinelibrary.wiley.com/doi/full/10.1029/2011JC007697>) and similarly, the NECC is also weakened around 6N in our simulation as shown in Fig.S6j. However, we acknowledge that a more in-depth analysis of the simulated patterns and modulations of the ocean current, and the subsequent impact on the surface freshwater budget, is beyond the scope of this study and needs to be investigated in future works.

In the revised manuscript, we have added more statements emphasizing the importance of the Amazon River plume, and the previous sentence at line 105 and reference #25 have been removed. We have added these discussions in the revised manuscript (please see lines 104-112, 153-160, and 212-213).

Both in the responses to the reviewers' comments and in the main text, the authors claim their study is the first one partially unveiling the impact of the Atlantic Niños over sea-air CO2 fluxes in the basin:

Response to reviewers: "We believe that our findings can be a good milestone for further investigations on CO2 flux response in different types of the Niño events since this study is the first study to reveal the CO2 flux response"

Lines 71-75 in the manuscript: "According to previous studies (8, 17-20), the Pacific and Atlantic Niños variability have a few different mechanisms, patterns and periodicities (e.g., ENSO Modoki, canonical Atlantic Niño, and early/late onset of Atlantic Niños). As a first study, we focus on the climatological peak of the Pacific and Atlantic Niños in November-December and June-July, respectively and investigate the most common CO2 flux response to the mean climate variability modes without any detailed categorization of events."

Lines 105-108: "Because there is no observational data with sufficient spatial and temporal coverage required to investigate the drivers of pCO2 change associated with the Atlantic Niño, the simulation of NorESM2 will be used to elucidate the driving mechanisms of the sea-air CO2 flux anomalies."

Lines 237-238: "Here, we have, for the first time, revealed the unique pattern and underlying mechanism of CO2 flux anomalies triggered by the Atlantic Niño"

While I recognize the value of this study, the authors should be aware of that three previous observational-based studies have studied the impact of an Atlantic Niño event over the sea-air CO2 exchange in the basin:

Lefèvre, N., Caniaux, G., Janicot, S., Gueye, A.K., 2013. Increased CO₂ outgassing in February-May 2010 in the tropical Atlantic following the 2009 Pacific El Niño. *Journal of Geophysical Research: Oceans* 118, 1645–1657. <https://doi.org/10.1002/jgrc.20107>

Ibánhez, J.S.P., Flores, M., Lefèvre, N., 2017. Collapse of the tropical and subtropical North Atlantic CO₂ sink in boreal spring of 2010. *Scientific Reports* 7, 41694. <https://doi.org/10.1038/srep41694>

Lefèvre, N., Veleda, D., Tyaquiçã, P., Perruche, C., Diverrès, D., Ibánhez, J.S.P., 2019. Basin-Scale Estimate of the Sea-Air CO₂ Flux During the 2010 Warm Event in the Tropical North Atlantic. *Journal of Geophysical Research: Biogeosciences* 124, 973–986. <https://doi.org/10.1029/2018JG004840>

The domain of these studies is different to that presented here and they are limited to a particular event, but they offer valuable observation on factors that I feel are not properly addressed by their model such as the abnormal displacement of the ITCZ during this event (Lefèvre et al., 2013), identifying warming as the main driver of the CO₂ anomalies observed (Ibánhez et al., 2017) and quantifying the impact over the sea-air CO₂ exchange in the basin north to the equator (Ibánhez et al., 2017, Lefèvre et al., 2019).

Other apparent limitations based on the results presented (for instance accounting for the anomalous displacement of the ITCZ, never mentioned in the manuscript) makes me feel that authors should improve the discussion of the limitations of the model while further mild the conclusions taken from their modeling results.

REPLY: Thank you very much for providing additional information and relevant references. First, in the previous version of the manuscript, we acknowledged Lefèvre et al. (2013), but we missed citing that paper in the previous manuscript. According to Lefèvre et al. (2013), the positive CO₂ flux anomaly associated with the warm SST anomaly in 2010 occurred in the north tropical Atlantic (note that their region of interest is slightly different from ours) occurring from February to May. This warm SST anomaly seems to be related to the Atlantic Meridional Mode (AMM) that has a peak during boreal spring (March to May). On the other hand, our focused SST anomaly is associated with the Atlantic Niño/Niña, which is also known as the Atlantic “Zonal” Mode (AZM) that has a peak, in general, during summer (June-July), and as we investigate in the manuscript. Our focus is also on the equatorial region, and we have made this clearer by changing “tropical Atlantic” to “equatorial Atlantic” through the text.

While some literatures suggest the connection between the North tropical/subtropical SST anomaly in spring and the equatorial Atlantic warm SST anomaly in summer (e.g., Martín-Rey and Lazar, 2019) through Rossby/Kelvin wave dynamics, the fundamental mechanisms generating these two modes are different, WES (wind-evaporation-SST) feedback for the AMM and Bjerknes feedback (wind, Kelvin wave, thermocline, and SST) for the AZM. Therefore, we believe that our findings have still novelty with respect to Lefèvre et al. (2013) and the subsequent studies (Ibánhez et al., 2017; Lefèvre et al., 2019).

However, we agree that it is useful to mention the pioneering results of Lefèvre et al. (2013) that show temperature and salinity play an important role in modulating the sea-air CO₂ flux, similar to our findings on the Atlantic Niño. We have added more discussion referring to these papers and descriptions on the simulated precipitation anomaly. Please see lines 147-149 and 216-221.